# Variability of Body Build and Physiological Spinal Curvatures of Young People in an Accelerated Longitudinal Study

**DOI:** 10.3390/ijerph18147590

**Published:** 2021-07-16

**Authors:** Jacek Tuz, Adam Maszczyk, Anna Zwierzchowska

**Affiliations:** 1Faculty of Health Sciences, Medical University of Silesia, 40-055 Katowice, Poland; 2Academy of Physical Education in Katowice, 40-959 Katowice, Poland; a.maszczyk@awf.katowice.pl (A.M.); a.zwierzchowska@awf.katowice.pl (A.Z.)

**Keywords:** variability of body build, variability of body posture, spinal curvatures in the sagittal plane

## Abstract

The human body is characterized by the variability of the characteristics of body build, which is expressed in the instability of spinal curvatures, which change during ontogeny. This phenomenon leads to a phylogenetic evolution of the human body build and posture. The aim of the study was to assess the dynamics of the variability of traits and indices of body build and posture and their mutual relations. It was assumed that over the 10-year observation period, a significant variability and relationships will be observed between the characteristics of body build and posture in young adults. Between 2006 and 2016, 2154 women and men aged 19.6 ± 0.8 were examined (first-year students at the University of Economics in Katowice, Poland). Measurements of basic anthropometric measurements and angular values of lumbar lordosis and thoracic kyphosis were performed. The collected data were analysed statistically, and the intergroup and intragroup differences were evaluated (ANOVA with repeated measures). The dynamics of variability (by building time series) and absolute and single-base increments were determined. The body build of young men and women in the period of ontogenetic stabilization in the study group has changed statistically significantly over the decade. A prediction of the increases in body weight and hip circumference was recorded over a period of 10 years, especially in men. The characteristics of body posture have also undergone a significant progressive change. In women, thoracic kyphosis increased by 96.15%, whereas in men, lumbar lordosis decreased by 52.65%. Significant sexual differentiation was shown for such characteristics as body height, waist circumference and the angle of lumbar lordosis. The relationships between the characteristics of body build and posture were verified. There was a moderate relationship between the hip circumference and lumbar lordosis in women and a weak relationship between body height and the angle of thoracic kyphosis in men.

## 1. Introduction

The human body, including the spinal curvatures, changes during ontogeny and phylogeny [1,2], which is reflected by the instability of characteristics and indicators of body build and posture. The period of ontogenetic stabilization between 20 and 40 years of age, accompanied by the inhibition of growth processes and establishment of body proportions does not guarantee discontinuation of the variability of physiological spinal curvatures. This is especially due to lifestyles, which are an important factor in changing body composition and body build in the ensuing decades. In this context, the phylogenetic changes observed in the increasing body weight and reducing body height during ontogenesis cannot be ignored [3]. This evolution is evidenced by the results of epidemiological studies, and their authors often indicate in their conclusions further consequences to the structure and functions of the human body, which are conducive to the occurrence of cardiometabolic diseases and joint and spine pain or they are looking for better and better ways of predicting them [4]. Other reports about being overweight or obese in early adulthood, as well as during the course of life generally, increases the risk of radiating but not non-specific LBP among men. “Taking into account the current global obesity epidemic, emphasis should be placed on preventive measures starting at youth, and also measures for preventing further weight gain later in life should be implemented” [5]. A meta-analysis demonstrates a strong relationship between LBP and decreased LLC, especially when compared with age-matched healthy controls. Among specific diseases, LBP by disc herniation or degeneration was shown to be substantially associated with the loss of LLC [6]. Back pain is common in middle-aged women. Increased weight, weight gain, and depression were independent predictors of back pain over 15 years, whereas participation in vigorous physical activity was protective. Targeting these lifestyle factors is an important area for future research on reducing the burden of back pain in middle-aged women [7]. Furthermore, it has been proven that there are endogenous posture compensation mechanisms that are a response to the evolution of body build. This process is not always associated with the symptoms of disease [8]. The human body has great adaptability, especially when the phylogenetic changes observed are spread over time. In this context, changes in physiological spinal curvature as seen from thorough observations, particularly when they contribute to abnormal posture and/or pain, are the subject of much research [9]. At the same time, physiological spinal curvatures are highly labile, which is recorded even in daily cycles, and the authors of these observations indicated their dependence on the mental status, lifestyle, and temporal pathologies that may also determine these characteristics. Body height decreases throughout the day due to fluid loss from the intervertebral disk [10] and multiple regression analysis indicates that height and the mean daily maximum temperature in August were statistically significant predictors of weight [11]. Low body mass index (BMI) is a well-established risk factor for fracture in postmenopausal women. Height and obesity have also been associated with increased fracture risk at some sites [12]. In these considerations, it is worth highlighting the focus on the symptoms and not on the tissue pathologies [13].

A disorder of the spatial system of the body posture may generate a state of spinal pathology, which is not always of a direct character. The sagittal balance is based on lumbar lordosis [14], with its decrease and/or increase considered to be the event initiating the cascade of compensatory mechanisms [15], which is also of a pathological nature. Furthermore, it is the part of the spine that is most often the target of surgical and physiotherapeutic interventions [16,17]. Asthenia of body build, being overweight, obesity, reduced muscle tension, and disproportionately high body fat compared to muscle mass are only some of the factors that coexist in the formation of spinal pathology [18].

The relationships of the physiological spinal curvatures with body build and body composition have been an area of exploration in a few scientific studies of phylogenetics, especially with relation to the young people in the period of ontogenetic stabilization.

The aim of this study was to evaluate the dynamics of the variability of characteristics and indices of body build and posture and their mutual relations. It was assumed that over the 10-year observation period, a significant variability and relationship between the characteristics of body build and posture will be observed in young adults.

## 2. Material and Methods

### 2.1. Participants

The examinations were carried out between 2006 and 2016. They were first year students (*n* = 2154) at the Faculty of Economics and Management of the local University of Economics. In total, 1314 women aged 19.7 years (±0.4) and 840 men aged 19.6 years (±1.2) were examined (see Table 1).

The students surveyed, representing the average young Caucasian population, were people who were involved in various forms of physical activity, but none of them was actively involved in sport. Furthermore, during the research, they did not report any pain or other diseases that would make it impossible to conduct the research and disturb the homogeneity of the group studied. The inclusion criterion concerned persons who had any locomotor system dysfunction or another disease preventing them from active participation in physical education classes in a university. The examinations were carried out in the morning in separate rooms on the premises of the university, and they were always carried out by the same two examiners. The subjects were wearing sportswear, without shoes and socks. Beforehand, they were instructed about the purpose of the study and fasted for 2 h before the measurements.

The research was carried out for the following 10 years at the same university (every year in October) and the same research procedures were used for first-year students. The research was conducted in accordance with the standards contained in the 2008 Declaration of Helsinki, and the students who participated in the examinations gave their written consent. Study participants could withdraw from the test procedure at any time without giving reasons. The first two years of the research were a pilot study and the data collected are included in the analyses of the research project “Diagnostics of the body build, body posture and physical fitness of students”, which was approved by resolution of the local bioethics commission No. 5/2008 on 29 April 2008.

### 2.2. Measurement Procedures

#### Procedures

The examinations were conducted in the morning. This was a cohort-sequential design cross-sectional analytical study, and the participants were measured only once. Anteroposterior spinal curvatures were evaluated by means of the Rippstein plurimeter. The plurimeter allows for quick, accurate, and inexpensive examinations of children and adolescents’ posture in the sagittal and transverse planes in order to complement physical examinations (in, e.g., orthopaedics and paediatrics) and rehabilitation. The apparatus enables quick and easy measurements and ensures the reproducibility of the results, even when the examinations are carried out by different examiners. Two values of angular deflection are obtained (read directly from the apparatus): the angle of thoracic kyphosis, measured between the kyphosis peak Th12 and Th1, and the angle of lumbar lordosis, measured between L5 and Th12. A V-plurimeter was employed to measure thoracic kyphosis and lumbar lordosis, with the patient standing without any postural correction. Two examiners performed the measurements in order to minimize inter-observer differences, see Figure 1. The value of 30 ± 5 was considered normal kyphosis and lordosis [9].

We measured body height (BH) and waist and hip circumference (WC and HC) [cm]. Body weight (kilograms) was determined using a Tanita BC 420SMA stand-on bioimpedance analyser (Tanita Corporation, Japan). The study was performed according to a standard protocol recommended by the manufacturer. The participant started examinations in the fasted state, in light clothing, shoeless, and without socks, with clean feet. The participants were instructed to fast for at least 2 h before the measurement. The device samples periods of 5-s resistance values (Rx) and the reactance of their volume (Xc). These data are used as a basis for calculating body composition by a computer program relative to age, sex, and body height [9]. Body height measurements were performed using a wall-mounted stadiometer with standard scales and an accuracy of 0.5 cm. Body height was measured to the nearest mm. An anthropometric tape was used over light clothing to measure waist circumference (WC [cm]) and hip circumference (HC [cm]). Waist circumference was measured between the iliac crest and the rib cage (minimum circumference) whereas hip girth was measured over the greater trochanters (maximum width).

### 2.3. Statistical Procedure

The collected material was subjected to statistical analysis. Arithmetic means and standard deviations were calculated. The Shapiro–Wilk test was used to test the data for normal distribution, whereas the homogeneity of variance was evaluated by means of the Levene’s test. The ANOVA analysis of variance with repeated measures was employed to determine intergroup and intragroup differences. Time series were built and absolute and single-base increments as well as single-base indices and variables were determined in order to assess the dynamics of the variability of specific variables. The time series were also used to determine trends and predictive values for lumbar lordosis and thoracic kyphosis. Correlations between the analysed variables were calculated using the Spearman’s test. The level of statistical significance was set at *p* < 0.05.

## 3. Results

It was demonstrated that the gender factor influences dependent variables such as body height, body weight, waist circumference, and lumbar lordosis at a level of statistical significance (*p* < 0.05) (see Table 2).

The analysis of the absolute values of variability over 10 years of observation of the characteristics of body build and body posture in both groups revealed the highest variability for thoracic kyphosis in both groups, and the lowest for body height in both men and women.

An increasing tendency for body height was observed in men, with a simultaneous decrease in hip and waist circumference. Different values were obtained for women, with body height tending to decrease with a simultaneous increase in hip and waist circumferences (Table 3).

The analysis of relative increments based on time series of observed characteristics of body build and posture in men allowed for the indication of the largest and smallest changes over the decade. Over 10 years of observation, of all the assessed characteristics of body build and body posture of the men studied, only the angle of kyphosis accelerated each year, and the mean value of development of this characteristic in this period was the highest (KTH10 = 52.65%), and was statistically significant. Furthermore, the highest range of variation among the somatic characteristics was found for hip circumference (HC10 = 30.64%). The smallest range of variation occurred for body height (BH10 = 3.13%).

The analysis of relative increments based on time series in women reveals that the greatest changes over the 10-year observation are observed for the angle of thoracic kyphosis (KTH10 = 96.15%) and body weight (BM10 = 14.37%).

The smallest changes during the 10-year observation in women were recorded for body height (BH10 = 3.32%). The variability of both the angle of kyphosis and lordosis were statistically significant (Table 4).

Analysis of single-base indices in men revealed slight changes in the characteristics of body build. The greatest variability of somatic characteristics was observed in the waist circumference (−2.3%), but it was not statistically significant. Similar significant changes, but of an accelerating nature, were recorded for the angle of thoracic kyphosis where the single-base index was 32.8% (Table 5).

The single-base indices in women were also analysed. The greatest variability was recorded in the waist circumference (10.7%). The index for the characteristics of body posture shows similar dynamics of variability in the case of thoracic kyphosis (58.9%). (Table 6).

A moderate relationship between body height and the angle of thoracic kyphosis was recorded in men and it was directly proportional (the higher the body height the higher the angle of thoracic kyphosis). Furthermore, a high and statistically significant relationship was observed between the hip circumference and lumbar lordosis in women. This relationship is directly proportional: the larger the hip circumference, the greater the lumbar lordosis (see Table 7). The other assessed characteristics of body build in both groups did not show such a significant relationship with respect to body posture.

## 4. Discussion

The assessment of the secular trend of overweight and obesity in school children was carried out between 1998 and 2008 in the south of Poland by Mazur et al. (a cross-sectional study) [19]. Significant shifts in the incidence of obesity and being overweight were found (a decrease in the incidence of obesity and an increase in the incidence of being overweight) among boys and girls in comparative studies in 1998 and 2008. Similar dynamics of the variability of the characteristics of body build (body height, body weight) were not observed in the presented studies of adults from southern Poland, although the presented studies were collected in subsequent years every year between 2006 and 2016 (single-base index, see Table 5 and Table 6).

The concurrent study of Malina provided a small longitudinal component [20], but the results failed to refer to the accelerated longitudinal methodology. Selected characteristics of body build were used only to assess the incidence of being overweight and obesity. Furthermore, Lehman et al. [21] analysed temporal changes and secular trends in body height and body weight in German conscripts in an accelerated longitudinal study. Although the research tools were similar, the objectives were already significantly different from those presented in our study. However, the authors of that study stressed the fact of non-linear variability of trends in the groups studied, which is consistent with our results of the assessed characteristics of body build and posture and the computed indices. Similar variability was demonstrated in a Brazilian study in a similar age group [22].

Observations of the secular trend of Caucasian, Asian, and African American populations clearly indicated a significant increase in body weight, with a simultaneous tendency to reduce body height regardless of the age groups analysed and the methodology used [23]. The lack of the increase in body height with the simultaneous increase in body weight and acceleration of puberty indicates progressive developmental disharmony [24]. The authors usually identify this fact with socio-economic factors, because the higher the economic status, the more sedentary people’s lifestyles and the more dynamic indicated trends are. Analysis of the dynamics of time series variability for the characteristics body build and posture also supported these theses as we noticed significant differentiation of single-base indices (Ij) by gender (Ij women: BH = 0.2%; BM = 7.9%), (Ij men: BH = 1.3%; BM = 1.2%). Our research supports such statements since both the physical activity and the physical fitness of Polish students has a downward trend and the economic status reveals an upward trend in such professions as economists, lawyers, managers, engineers, and healthcare workers. The studies showed a relationship between physical activity and the socio-economic status of the respondents [25].

Furthermore, such a significant variation in the variability of somatic characteristics observed in relation to gender was confirmed in a study by Tutkuviene et al. in 2005, who pointed out that it is a secular trend in women that is particularly noticeable [26]. Our research supports this thesis as the relative mean increase in body mass over 10 years in the observed homogeneous population of young women (3.62 kg) compared to men (1.81 kg) was significantly higher.

The changes in body build observed in our study are not as impressive as in the case of the characteristics of body posture, where relative increases based on chain indices were KTH = 52.65% and LL = 36.01% for men and KTH = 96.15% and LL = 49.2% for women. Unfortunately, the lack of similar observations in the available literature does not allow for a broader interpretation of the results. However, the observed variability of the angles of kyphosis and lordosis is statistically significant and much higher compared to somatic characteristics. This fact already confirms how plastic and sensitive the skeletal system of the spine is to changes in body build.

Malina et al. and Lehman et al. [20,21] stressed the importance of the identification of risk factors contributing to being overweight or obese. However, none of the researchers predicted the effect of variability of BMI (increase) on the physiological spinal curvatures, which may result in pathologies. Lehman and Avilla [21,22] evaluated only men. However, a high correlation is observed between the increase in hip circumference and lumbar lordosis in women and a moderate correlation between body height and thoracic kyphosis in men, which in turn affects the occurrence of compensatory changes in the spine and joints [8].

In conclusion, significant sexual differentiation was shown for such characteristics as body height, waist circumference, and the angle of lumbar lordosis. Body build and body posture of young people in a longitudinal observation over the decade have changed significantly with an unfavourable tendency to increase body weight. During the 10-year observation, a significant decrease in lumbar lordosis was observed, with the simultaneous deepening of thoracic kyphosis.

### Strengths and Weaknesses of the Study

The main limitation of the study is the difficulty of discussing the results, as similar relationships have been shown only in causal studies in different groups according to age, physical activity, and gender.

The absolute strength of the presented results is their continuous character, methodological repeatability, and the fact that the empirical research was carried out over a period of 10 years by the same three examiners. This is of particular importance when concluding on the reliability of physiological measurements of spinal curvature.

## 5. Conclusions

The characteristics of human body build and posture during the period of ontogenetic stabilization show high non-linear variability, characterized by significant dimorphic differentiation. A moderate relationship was found between the hip circumference and lumbar lordosis in women, whereas a weak relationship occurred between body height and the angle of thoracic kyphosis in men.

## Figures and Tables

**Figure 1 ijerph-18-07590-f001:**
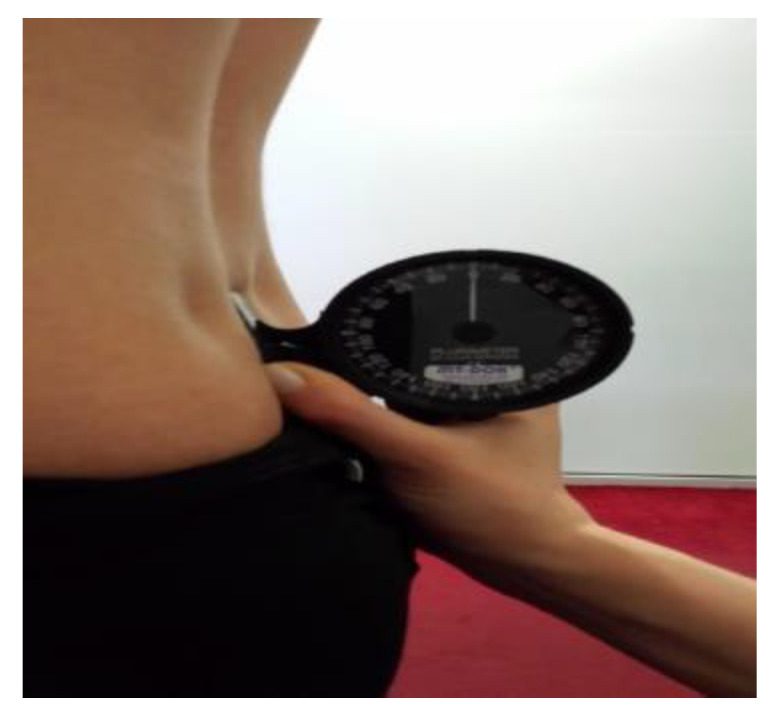
Own source.

**Table 1 ijerph-18-07590-t001:** Structure of the study group of women and men in the period between 2006 and 2016.

Date of Examination	Women			Men		
	*n*	Age S ± SD	Min–Max	*n*	Age S ± SD	Min–Max
2006	139	19.3 ± 0.5	17.8–20.7	119	19.4 ± 1.2	18–24
2007	230	19.7 ± 0.77	17.4–22	152	19.7 ± 1.01	18–23
2008	131	19.5 ± 0.67	17.8–23.2	99	19.1 ± 0.6	18–22
2009	99	19.9 ± 0.9	18.2–24.5	40	19.6 ± 1.1	18–24
2010	167	19.6 ± 0.6	18.3–23.1	89	19.5 ± 1.04	19–22
2011	163	19.1 ± 0.6	18–22	94	19.2 ± 1.06	18–24
2012	137	19.1 ± 0.7	18–23	71	19.1 ± 0.8	18–23
2013	62	19.6 ± 0.7	18.2–22.1	23	20.3 ± 1.1	19–23
2014	59	20.1 ± 0.6	18.8–20.9	72	19.1 ± 0.9	18–23
2015	75	20.3 ± 1.0	17.7–24.8	52	20.4 ± 0.8	18–23
2016	52	20.3 ± 0.6	18.7–21.5	29	20.1 ± 1.1	17–23
Total	1314	19.7 ± 0.4	18.1–22.5	840	19.6 ± 1.2	18.2–23.5

**Table 2 ijerph-18-07590-t002:** Significant gender differences in the analysed characteristics of body build and posture between women and men in 2006–2016.

Characteristic	Men (*n* = 840)		Women (*n* = 1314)		F	*p*
	$\bar{x}$	Min.–Max.	$\bar{x}$	Min.–Max.		
BH [cm]	180.6 ± 6.3	161–202	168 ± 6.1	151.0–189.9	444.498	0.001 *
BM [kg]	74.9 ± 12.4	49.3–139	58.7 ± 8.8	39.4–108.5	131.066	0.001 *
WC [cm]	82 ± 8.8	52–118	70.9 ± 6.4	59.0–112.0	44.299	0.001 *
HC [cm]	80 ± 8.8	70 –91	95.7 ± 6.6	78.0–132.0	0.375	0.544
KTH [°]	37.5 ± 8.5	9–64	35.7 ± 8.5	15.0–62.0	0.406	0.531
LL [°]	30 ± 9.4	4–60	34.4 ± 8.6	10.0–60.0	6.774	0.002 *

* statistical significance at *p* < 0.05, BH—body height, BM—body mass, WC—waist circumference, HC—hip circumference, KTH—thoracic kyphosis, LL—lumbar lordosis.

**Table 3 ijerph-18-07590-t003:** Increases in absolute values of the body build and body posture of men and women in 2006–2016.

Participants	BH [cm]	BM [kg]	HC [cm]	WC [cm]	KTH [°]	LL [°]
Women	−0.37	3.62	1.14	2.96	46.87	14.64
Men	0.23	1.81	−2.81	−1.37	43.26	12.65

**Table 4 ijerph-18-07590-t004:** Relative increments based on chain indices for body posture and body build in men and women [%].

Characteristics of Body Build and Body Posture	Men			Women			
	Min.	Max.	$\bar{x}$ _10_	Min.	Max.	$\bar{x}$ _10_	*p*
BH	−1.60	1.53	3.13%	−2.14	1.18	3.32%	0.676
BM	−10.13	6.84	16.97%	−3.40	10.97	14.37%	0.545
WC	−6.76	11.25	18.01%	−2.20	11.39	13.59%	0.051
HC	−24.15	6.49	30.64%	−4.57	5.89	10.46%	0.001
LL	−6.00	30.01	36.01%	−18.25	30.95	49.20%	0.002
TH	32.30	84.95	52.65%	−5.50	90.65	96.15%	0.001

**Table 5 ijerph-18-07590-t005:** Dynamics of the variability of time series for the characteristics of body build and posture in men [%] in terms of single-base indices (Ij) and variable-base indices (Iz) between 2006 and 2016.

Characteristics of Body Build and Body Posture	Variable-Base Index (Iz) Min. [%]	Variable-Base Index (Iz) Max. [%]	Single-Base Index (Ij) Until 2006 [%]
Body height	2012 to 2011 −3.8	2013 to 2012 4.7	1.3
Body mass	2013 to 2012 −29.5	2012 to 2011 26.8	1.2
Waist circumference [cm]	2012 to 2011 −27.2	2011 to 2010 24.3	−2.3
Hip circumference [cm]	2013 to 2012 −51.6	2012 to 2011 62.9	−2.1
LL	2015 to 2014 −57.5	2014 to 2013 50.2	−16.5
Th	2011 to 2010 −37.1	2007 to 2006 32.3	32.8

**Table 6 ijerph-18-07590-t006:** Dynamics of variability of time series for the characteristics of body build and posture in women [%] in terms of single-base indices (Ij) and variable-base indices (Iz) between 2006 and 2016.

Characteristics of Body Build and Body Posture	Variable-Base Indices (Iz) Min. [%]	Variable-Base Index (Iz) Max. [%]	Single-Base Index (Ij) Until 2006 [%]
Body height	from 2012 to 2011 −3.5	from 2013 to 2012 3.2	−0.2
Body mass	from 2013 to 2012 −15.3	from 2012 to 2011 15.2	7.9
Waist circumference [cm]	from 2013 to 2012 −12.4	from 2012 to 2011 10.2	10.7
Hip circumference [cm]	from 2013 to 2012 −17.5	from 2014 to 2013 18.3	1.9
LL	from 2015 to 2014 −75.1	from 2014 to 2013 90.7	15.1

**Table 7 ijerph-18-07590-t007:** Correlations of the variables of body build and posture in men and women in a 10-year observation.

	Women		Men		*p*-Value Kyphosis	*p*-Value Lordosis
Variable	Angle of Kyphosis	Angle of Lordosis	Angle of Kyphosis	Angle of Lordosis		
Height	0.30	0.1	0.40	−0.06	0.057	0.001 **
Body mass	0.20	0.32	−0.20	−0.04	0.002 **	0.001 **
Waist circumference [cm]	0.30	0.13	−0.20	0.13	0.001 **	0.974
Hip circumference [cm]	0.10	0.70	−0.10	0.08	0.001 **	0.001 **

** statistically significant differences women/men.

## Data Availability

Please contact with autors for data requests.

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
