# Peer review of "Variability of Body Build and Physiological Spinal Curvatures of Young People in an Accelerated Longitudinal Study"

_ijerph, 2021, doi:10.3390/ijerph18147590_

Round 1

Reviewer 1 Report

In this paper authors presented: Variability of body build and physiological spinal curvatures of young people in an accelerated longitudinal study.

The following aspects should be improved:

  • It is unclear which angles of lordosis and kyphosis are considered? How they were measured? Internal spinal alignment, or external asymmetry line was considered?
  • Figure of a subject with detailed annotations is missing.
  • Here, only normal/physiological spinal curvature in sagittal plane is analyzed? What if subject has some spinal deformity? It is not clear what were inclusion / exclusion criteria.
  • Refs 4,5,6,7 and 10,11,12,13 should be elaborated in more details.
  • On Page 4, there is a reference to a Fig.1 , but this figure doesn’t exist in the manuscript.
  • Figure about setup and measurement is missing also for the Rippstein plurimeter. It is hard to get impression about what and how measurements were collected from subjects.
  • If only sagittal profile is analyzed, why was cervical lordosis avoided?
  • Please check the manuscript against typos.
  • On Page 7, brackets are empty “… an upward trend ().”

Author Response

Jacek Tuż

Reviewer 2 Report

The paper is well written and all the sections are well described. I consider that the current study is relevant and of general interest to the readers of the journal. Below are my comments to the authors:

Page 4: The authors state: ‘Two examiners performed the measurements in order to minimize interobserver differences, see Fig. 1.’, but Figure 1 is missing in the article.

Page 5: Please add at the end of the first and second paragraph (Tab. 4). You should add also the p-value in the text any time you state that the differences are statistically significant or add a column in the table with the p-values.

Page 6: You should add the p-value in the text when you state that the differences are statistically significant or add a column in the table with the p-value. In Table 7 is indicated with a symbol that the correlation is significant but the p-value is missing.

Page 7, Discussion section, third paragraph: In the sentence ‘Our research supports such statements since both the physical activity and the physical fitness of Polish students has a downward trend and the economic status reveals an upward trend ( )’ it seems that the reference is missing.

Results concerning statistical analysis are not fully described. When the difference/correlation is stated to be significant the p-value should be added to the manuscript text in brackets or add the exact p-value in the corresponding table. I would encourage the authors to add a column in each table with the p-values when it applies as it is shown in Table 2.

Author Response

Jacek Tuż

Round 2

Reviewer 1 Report

N/A